# Effects of Different Rare Earth Elements on the Degradation and Mechanical Properties of the ECAP Extruded Mg Alloys

**DOI:** 10.3390/ma15020627

**Published:** 2022-01-14

**Authors:** Junxiu Chen, Jie Zhan, Sharafadeen Kunle Kolawole, Lili Tan, Ke Yang, Jianhua Wang, Xuping Su

**Affiliations:** 1Key Laboratory of Materials Surface Science and Technology of Jiangsu Province, Changzhou University, Changzhou 213164, China; zhanjiechn@yeah.net (J.Z.); wangjh@cczu.edu.cn (J.W.); sxping@cczu.edu.cn (X.S.); 2Jiangsu Collaborative Innovation Center of Photovoltaic Science and Engineering, Changzhou University, Changzhou 213164, China; 3National Experimental Teaching Demonstration Center of Materials Science and Engineering, Changzhou University, Changzhou 213164, China; 4Mechanical Engineering Department, School of Engineering and Technology, Federal Polytechnic, Offa P.M.B 420, Nigeria; eskaypumping@yahoo.co.uk; 5Institute of Metal Research, Chinese Academy of Sciences, Shenyang 110016, China; lltan@imr.ac.cn

**Keywords:** ECAP, rare earth elements, Mg alloy, corrosion resistance, mechanical properties

## Abstract

Effects of different rare earth elements on the degradation and mechanical properties of the ECAP (equal channel angular pressing) extruded Mg alloys were investigated in this work. Microstructural characterization, thermodynamic calculation, a tensile test, an electrochemical test, an immersion test, a hydrogen evolution test and a cytotoxicity test were carried out. The results showed that yttrium addition was beneficial to the improvement of the alloy’s strength, and the ultimate tensile strength (UTS) and yield strength (YS) values of the ECAPed Mg-2Zn-0.5Y-0.5Zr alloy reached 315 MPa and 295 MPa, respectively. In addition, Nd was beneficial to the corrosion resistance, for which, the corrosion rate of the ECAPed Mg-2Zn-0.5Nd-0.5Zr alloy was observed to be 0.42 ± 0.04 mm/year in Hank’s solution after 14 days of immersion. Gd was moderate in improving both the corrosion resistance and mechanical properties. Moreover, after co-culturing with murine calvarial preosteoblasts (MC3T3-E1) cells, the ECAPed Mg-2Zn-0.5RE (Nd, Gd, Y)-0.5Zr alloys exhibited good cytocompatibility with a grade 1 cytotoxicity. Consequently, the ECAPed Mg-2Zn-0.5Nd-0.5Zr alloy showed the best application prospect in the field of orthopedics.

## 1. Introduction

Magnesium (Mg) and its alloys as biodegradable implants continue to gain prominence in the clinics because they have shown a promising application prospect in the biomedical field. However, the fast degradation rate and relatively low mechanical properties still limit their wide applications. Alloying and plastic deformation are two kinds of useful methods to improve both the mechanical and degradation properties of Mg alloys.

Plastic deformation enhances the degradation resistance by changing the distribution and morphologies of the second phases. After plastic deformation, and especially severe plastic deformation, the large and bulk second phases will be broken into small particles and distributed uniformly. Thus, the degradation resistance is enhanced. Lotfpour et al. [1] found that the corrosion rate of the Mg-2Zn-0.3Cu alloy decreased from approximately 9.0 mm/year (as-cast) to 7.0 mm/year after hot extrusion due to the redistribution of the intermetallics. Li et al. [2] studied the corrosion resistance of pure Mg, Mg-1Ca and Mg-2Sr alloys processed by ECAP. Compared to the as-cast counterparts, they found that the corrosion rate of pure Mg increased, whereas that of Mg-1Ca and Mg-2Sr alloys decreased. Further study showed that ECAP could decrease the corrosion rate of high-alloying Mg alloys and increase that of the relatively low-alloying Mg alloys [3]. Besides, the mechanical properties of the Mg alloys were also improved due to the grain refinement strengthening and second phase strengthening [4]. 

Rare earth (RE) elements, such as Gd, Nd and Y, are widely added into Mg alloys to enhance their inherent properties [5]. They exhibit excellent strengthening effects and degradation resistance. Gd, for instance, could stabilize the corrosion layer of Mg alloys. Moreover, Deepsaark et al. [6] found that an addition of 2 wt.% Gd could improve the elongation of the Mg-2Sr alloy by approximately twice as much. Nd, on the other hand, has good precipitation strengthening and solution strengthening properties. The strength of Mg-Nd-Zn-Zr alloys was increased to the range of 320–380MPa. Moreover, Nd has the ability to weaken the microgalvanic effect between the second phases and the matrix [7]. Likewise, Y could improve the degradation resistance by the formation of Y_2_O_3_ in the degradation layer of Mg alloys [8]. The respective solubilities of the Nd, Y and Gd in Mg alloys are 3.6 wt.%, 12.4 wt.% and 23.49 wt.% [9]. They represent the different kinds of RE-containing Mg alloys with different solubilities. These three different elements were thus chosen in this study. In addition, these three elements contained in alloys have been studied, such as the WE43 alloy, Mg-Nd-Zn-Zr alloy and Mg-Gd-Zn-Zr alloy [10]. Until now, few studies that detail the different roles of the three elements in the degradation and mechanical properties of the wrought Mg alloys are available. Therefore, the effects of the three rare earth elements on the degradation and mechanical properties of the ECAP extruded Mg alloys were studied in this work.

## 2. Experimental Details

### 2.1. Materials Fabrication

The composition of the three as-cast alloys is illustrated in Table 1. The bars with size of 10 × 10 × 100 mm were cut from the as-cast ingots, and were put into the extrusion mold (Figure 1a). The mold and the bars were heated at 380 °C for 1.5 h. Then, the bars were extruded four times. After each extrusion, the mold was open immediately and the bars were rotated 90° at the same direction. 

### 2.2. Microstructural Characterization

After a four-pass extrusion, the samples microstructures were observed by using optical microscope (OM, Japan, OLYMPUSGX71) and scanning electron microscope (SEM, America, Inspect F50). The texture of the extruded samples was analyzed with the aid of XRD. The second phases composition and the distribution were analyzed by using a transmission electron microscope (TEM, America, Tecnai G20).

### 2.3. Thermodynamic Calculation

In order to evaluate the chemical activity of the three rare earth elements, the thermodynamic calculation was carried out by using Factsage 8.0 software. Gibbs free energy (ΔG) of the reaction between the three elements and oxygen at 298.15 K and 1273.15 K was calculated. 

### 2.4. Mechanical Test

The tensile samples were fabricated according to Figure 1b. The test was performed at room temperature with an Instron-5569 universal testing machine (1.0 mm/min tensile rate). The tensile force and displacement curves were obtained. Three tests were carried out for each sample. After tensile test, the fracture morphologies were observed by using SEM.

### 2.5. Electrochemical Test

The samples (10 × 10 × 5 mm) were coated with epoxy resin, where one side of the sample was connected with copper wire, and the other side was exposed. The exposed side (10 × 10 mm) was polished with 2000-grit SiC sandpaper. After polishing, they were rinsed using anhydrous ethanol and then dried. The electrochemical experiment was carried out by using the electrochemical workstation (Reference 600) produced by Gamry Company. The typical three-electrode system was adopted for the test. The sample was the working electrode, the calomel electrode (SCE) was the reference electrode and the platinum plate electrode was the counter electrode. The electrochemical experiment was carried out in Hank’s solution at 37 °C. The composition of Hank’s solution is shown in Table 2. First, 1800s open circuit potential (OCP) test was carried out to ensure that the whole measurement system was in a stable state. Then electrochemical impedance spectroscopy (EIS) test was carried out, and the frequency range was set to 10^5^–0.01 Hz under OCP. Finally, the polarization curves were tested. With the open circuit potential as the reference, the initial potential of the polarization curve was −0.25 V, the terminal potential was 0.35 V and the scanning rate was 0.5 mV/s. In order to ensure the accuracy of the experimental results, each group of samples was repeated three times.

### 2.6. Immersion Test

The samples with size of 10 × 10 × 3 mm were cut using a line cutting machine. They were then polished with 2000-grit SiC sandpaper and subsequently immersed in Hank’s solution according to the immersion ratio of 1.25 cm^2^/^mL^. The pH value of the solution was adjusted to 7.40 before immersion. The solution was changed every day in order to keep it fresh, and the pH value was recorded every day. After 7 days and 14 days of immersion, the samples were taken out. The degradation rate (*P*) was calculated according to Equation (1) after the degradation products were cleaned by using chromic acid. The micro and macro structures of the samples were then observed.
(1)P=(K×Δm)/(A×T×D) (mm/yr)
where *K* is a constant with value of 8.76 × 10^4^, Δm is the sample weight loss (g), *A* is the sample surface area (cm^2^), *T* is the sample immersion time (h) and *D* is the sample density. 

### 2.7. Hydrogen Evolution Test

The hydrogen evolution installation is presented in Figure 2. The samples with size of 10 × 10 × 3 mm were placed in beakers containing Hank’s solution. In the experimental set-up, the funnel was placed over the samples to collect hydrogen from the sample surface. The burette was then installed over the funnel and Hank’s solution subsequently injected. With this method, the volume of the escaping hydrogen was measured. The corrosion rate (*CR_HE_*) was then calculated according to Equation (2) [11] after 14 days of immersion.
(2)$$CR_{HE}=\frac{24.31\times \Delta V_{H_{2}}\times 365\times 24\times 10^{-2}}{22.4\times A\times T\times \rho }=95\frac{\Delta V_{H_{2}}}{A\times T\times \rho }\left(mm/yr\right)$$
where $\Delta V_{H_{2}}$ is the hydrogen evolution volume during the 14 days of immersion (mL), *A* is the sample surface area (cm^2^), *T* is the hydrogen evolution time (h) and ρ is the density of the magnesium alloys (g/cm^3^).

### 2.8. Cytotoxicity

Murine calvarial preosteoblasts (MC3T3-E1) cells were used to investigate the cytocompatibility of the ECAPed samples. The cells were cultured in the Dulbecco’s modified Eagle’s medium (DMEM) with 1% streptomycin/penicillin and 10% (*v*/*v*) fetal bovine serum (FBS) in 5% CO_2_ humidified atmosphere (37 °C). First, the samples were immersed in a serum-free medium for 24 h at 37 °C according to the immersion ratio of 1.25 cm^2^/mL. Subsequently, a 0.22 μm filter was used to filter the collected supernatant. Cells with density of 1 × 10^4^ cells/mL were incubated in the 96-well plates overnight. Then, the medium was replaced with 100 μL of various extracts, respectively. The group with only DMEM medium was the negative control, whereas 10% DMSO was the positive control. After culturing for different times (1, 3, 5 days), 10 μL MTT was added into each plate well. They were then co-cultured at 37 °C for 4 h, and then 150 μL dimethylsulfoxide (DMSO) was added to each well. After mixing, 100 μL supernatant of each well was moved into a new 96-well plate and the optical density (*OD*) values were measured by using a nanodrop spectrophotometer (Eppendorf, German) at 490 nm with a reference wavelength of 570 nm. The cell relative growth rate (*RGR*) was obtained according to Equation (3) [12]
(3)$$RGR=OD_{test}/OD_{negative}\times 100\%$$

## 3. Results

### 3.1. Microstructural Analysis

Figure 3 shows the optical microstructures of the extruded samples. For the as-cast Mg-2Zn-0.5Zr alloy, there were almost no second phases observed in the grain boundaries. For the as-cast Mg-2Zn-xRE-0.5Zr alloys, there were some second phases distributed along the grain boundaries, especially in the Mg-2Zn-0.5Y-0.5Zr alloys. After the four-pass extrusion, bimodal grain structures were obviously observed. The grain sizes of the alloys are presented in Table 3. For the as-cast alloys, both Nd and Gd exhibited a good grain refinement; however, the Y element did not. After the four-pass extrusion, all of the alloy grain sizes decreased significantly. The grain sizes of the Mg-2Zn-0.5Zr alloy, Mg-2Zn-0.5Gd-0.5Zr alloy, Mg-2Zn–0.5Nd-0.5Zr and Mg-2Zn–0.5Y-0.5Zr alloy were 4.99 ± 2.20 μm, 4.40 ± 0.86 μm, 3.95 ± 0.26 μm and 2.84 ± 0.32 μm, respectively. Moreover, the extruded Mg-2Zn-0.5Y-0.5Zr alloy showed the most uniform and smallest grain size. Figure 4 shows the SEM images of the as-cast and extruded alloys. For the Mg-2Zn-0.5Zr alloy, there were only some small particles observed in the as-cast. After the four-pass extrusion, some Zr-rich second phases were observed. For the Mg-2Zn-0.5Gd-0.5Zr alloy, the second phases were mainly distributed along the grain boundaries with the formation of a network structure. After the four-pass extrusion, the network structure was broken and some second phase enrichment areas were observed. For the Mg-2Zn-0.5Nd-0.5Zr alloy, there were some dendritic second phases observed along the grain boundaries. After the four-pass extrusion, the dendritic second phases were broken and the second phases changed to a columnar structure. For the Mg-2Zn-0.5Y-0.5Zr alloy, the second phases were mainly rodlike and dendritic. After the four-pass extrusion, the second phases changed into small particles. Moreover, the small second phases were still distributed along the grain boundaries. Figure 5 presents the TEM images of the samples after the four-pass extrusion. Figure 5a,b show the microstructure of the Mg-2Zn-0.5Gd-0.5Zr alloy. According to the diffraction spots and after calculating the crystal plane space, it was inferred that the second phases were mainly the W phase (Mg_3_Gd_2_Zn_3_ phase) (Figure 5a). Besides the large second phases, some small second phases were also observed. According to a previous report, the small second phases were mainly Mg-Zn phases. Figure 5c,d show the microstructures of the Mg-2Zn-0.5Nd-0.5Zr alloy. According to the EDS analysis, the atomic ratio of Mg, Zn and Nd is 77:14:9, which is close to (MgZn)92.1RE7.9, namely, the T phase. Therefore, it can be inferred that the second phases were the T phase. Compared with the Mg-2Zn-0.5Gd-0.5Zr alloy, the amount of small second phases increased obviously. Moreover, the dispersive second phases were much smaller. Figure 5e,f shows the microstructures of the Mg-2Zn-0.5Y-0.5Zr alloy. According to the diffraction spots and after calculating the crystal plane space, it was inferred that the second phases were mainly the W phase (Mg_3_Y_2_Zn_3_ phase). The small second phases were more uniformly distributed in the alloy. According to the dispersion strengthening theory, it was inferred that the Mg-2Zn-0.5Y-0.5Zr alloy should have the highest strength. Figure 6 presents the textures in the four-pass extruded alloys. For the Mg-2Zn-0.5Zr alloy, the main texture was {$\bar{2}$110} and {10$\bar{1}0$}. After the addition of Gd, Nd and Y, the main texture was {10$\bar{1}0$}, whereas the {$\bar{2}$110} texture disappeared. The RE elements therefore changed the texture of the alloys. 

### 3.2. Thermodynamic Calculation

Table 4 shows the Gibbs free energy (ΔG) of the reaction between the rare earth elements and oxygen at 298.15 K and 1273.15 K, respectively. Generally speaking, the lower the ΔG, the easier the reaction that occurred, and the element therefore has more chemical activity. It can be found that the ΔG was ranked as $\Delta G_{Y}<\Delta G_{Gd}<\Delta G_{Nd},$ irrespective of the low temperature (298.15 K) or high temperature (1273.15 K). Therefore, it can be concluded that Y was more likely to be oxidized and Nd was not easily oxidized. 

### 3.3. Mechanical Properties

Figure 7 presents the stress–strain curves of the alloys after the four-pass extrusion. Table 5 illustrates the mechanical properties of the alloys after the four-pass extrusion. For the Mg-2Zn-0.5Zr alloy, no obvious yield platform and work-hardening process were observed. The ultimate tensile strength (UTS), yield strength (YS) and elongation (EL) were the lowest. The addition of the rare earth elements obviously improved the mechanical properties. The Mg-2Zn-0.5Nd-0.5Zr alloy showed the best plasticity, with an elongation of approximately 22%. The Mg-2Zn-0.5Y-0.5Zr alloy showed the highest strength, with UTS and YS values of 315 MPa and 295 MPa, respectively. Figure 8 presents the fracture morphologies of the alloys after the four-pass extrusion. For the Mg-2Zn-0.5Zr alloy, there were many cleavage steps observed and the numbers of dimples were relatively small, indicating that it was mainly a brittle fracture. For the other alloys, the number of dimples increased obviously, especially in the Mg-2Zn-0.5Nd-0.5Zr alloy, exhibiting a typical ductile fracture.

### 3.4. Electrochemical Tests

Figure 9 presents the potentiodynamic polarization curves of the four-pass extruded alloys. For potentiodynamic polarization curves, the more to the left means the slower corrosion rate. The Mg-2Zn-0.5Nd-0.5Zr alloy was the leftmost, indicating that it should have the best corrosion resistance. Table 6 shows the Tafel fitting results of the potentiodynamic polarization curves of the alloys. There are two important parameters that are closely related to the corrosion resistance, namely, the corrosion potential (*E*) and corrosion current density (*i_corr_*). Although the corrosion potential (*E*) is not closely related to the corrosion resistance, it still reflects the corrosion tendency. Therefore, it can be observed that the Mg-2Zn-0.5Nd-0.5Zr alloy had the least tendency to be corroded, with the smallest value of −1.47 ± 0.01 V. The corrosion current density (*i_corr_*) is a parameter that reflects the corrosion resistance directly. The higher the value of *i_corr_*, the faster the corrosion rate (*CR*). The *CR* was calculated by using Equation (4).
(4)$$CR=22.85i_{corr}$$

The corrosion rate was ranked as follows: Mg-2Zn-0.5Zr alloy > Mg-2Zn-0.5Y-0.5Zr alloy > Mg-2Zn-0.5Gd-0.5Zr alloy > Mg-2Zn-0.5Nd-0.5Zr alloy. The corrosion resistance was improved by at least two times after the addition of the rare earth elements. 

Figure 10 shows the EIS curves of the alloys after the four-pass extrusion operation. From the Nyquist curves observation, all the curves mainly contain two arcs, namely, the capacitance loop at a high frequency and inductance loop at a low frequency. The diameter of the capacitance loop reflects the corrosion resistance directly. The larger the diameter of the capacitance loop, the better its corrosion resistance. It can be seen that the Mg-2Zn-0.5Nd-0.5Zr alloy has the largest diameter, indicating the best corrosion resistance. In contrast, the Mg-2Zn-0.5Zr alloy has the smallest diameter, meaning the lowest corrosion resistance. Moreover, the inductance loop was closely related to the hydrogen evolution; that is, the larger the diameter of the inductance loop, the greater its hydrogen evolution ability. It can be seen that the Mg-2Zn-0.5Nd-0.5Zr alloy has only a small inductance loop, meaning that there was only a small amount of hydrogen evolved from the alloy surface. Figure 10b shows the relationship between the frequency and impedance modulus. The impedance modulus at a low frequency also reflects the alloy’s corrosion resistance. The larger the value of the impedance modulus, the better the corrosion resistance. It can be seen that the Mg-2Zn-0.5Nd-0.5Zr alloy has the maximum impedance modulus at a low frequency (0.01Hz); therefore, from the Bode curves analysis, it can also be concluded that the alloy has the best corrosion resistance. Figure 10c shows the relationship between the Bode phase angle and the frequency. There are mainly two wave crests. The wave crest at low and high frequencies represent the inductance loop and capacitance loop. Figure 10d shows the equivalent circuit of the EIS curves, where *R*_s_ is the solution resistance. The constant phase element *CPE*_1_ in parallel with the film resistance *R*_1_ was used to describe the capacitance loop at a high frequency. In addition, the constant phase element *CPE*_2_ in parallel with the charge transfer resistance *R*_2_ represents the capacitance loop at a medium frequency. An inductance *L* and *R_3_* represent the inductance loop at a low frequency. The *CPE* is a constant phase element that can replace an ideal capacitor to compensate for the non-homogeneity in the system. Table 7 shows the fitting results of the EIS curves based on the equivalent circuit. It can be found that the Mg-2Zn-0.5Nd-0.5Zr alloy has a much larger resistance *R* (*R*_1_ + *R*_2_ + *R*_3_) value and lower *CPE* (*CPE*_1_ + *CPE*_2_) value. According to previous reports [14], the larger *R* value means a lower dissolution rate of the sample, and the lower *CPE* value signifies a compact surface. Therefore, it can also be concluded that the Mg-2Zn-0.5Nd-0.5Zr alloy has the best corrosion resistance.

### 3.5. Immersion Tests

Figure 11a presents the hydrogen evolution properties of the samples in Hank’s solution. It can be found that the total amount of hydrogen evolved from the Mg-2Zn-0.5Zr, Mg-2Zn-0.5Y-0.5Zr, Mg-2Zn-0.5Gd-0.5Zr and Mg-2Zn-0.5Nd-0.5Zr alloys was approximately 15.0 mL, 12.0 mL, 9.0 mL and 6.5 mL, respectively. The more the hydrogen evolved from the samples, the faster the corrosion rate. The corrosion rate after 14 days of immersion can be calculated according to a combination of the reaction ($\mathrm{Mg}+2H_{2}O\to Mg{\left(OH\right)}_{2}+H_{2}↑$) and Equation (2). Therefore, the corrosion rates of the Mg-2Zn-0.5Zr, Mg-2Zn-0.5Y-0.5Zr, Mg-2Zn-0.5Gd-0.5Zr and Mg-2Zn-0.5Nd-0.5Zr alloys were 0.86 mm/y, 0.69 mm/y, 0.52 mm/y and 0.37 mm/y, respectively. Figure 11b shows the pH change in the solution during the immersion period. The Mg-2Zn-0.5Nd-0.5Zr alloy also shows the lowest pH value. Figure 11c presents the corrosion rate of the samples after 7 days and 14 days of immersion in Hank’s solution. The corrosion rates of the Mg-2Zn-0.5Zr, Mg-2Zn-0.5Y-0.5Zr, Mg-2Zn-0.5Gd-0.5Zr and Mg-2Zn-0.5Nd-0.5Zr alloys after 7 days of immersion were 0.77 ± 0.10 mm/y, 0.63 ± 0.06 mm/year, 0.48 ± 0.09 mm/year and 0.35 ± 0.05 mm/y, respectively. After 14 days of immersion, the corrosion rates of all of the samples increased slightly, recording 0.80 ± 0.10 mm/y, 0.65 ± 0.05 mm/year, 0.53 ± 0.06 mm/year and 0.42 ± 0.04 mm/year, respectively. Figure 12 shows the corrosion morphologies of the four-pass extrusion alloys after 14 days of immersion in Hank’s solution with corrosion products. It was found that there were many corrosion products deposited on the surface. However, the corrosion products on the surface of the Mg-2Zn-0.5Nd-0.5Zr alloy were more compact and exhibited a cluster appearance. According to the EDS analysis (Table 8), it was observed that the clusters were mainly MgO and calcium phosphates. Figure 13 shows the corrosion morphologies of the four-pass extrusion alloys after 14 days of immersion in Hank’s solution with the removal of the corrosion products. For the Mg-2Zn-0.5Zr alloy, it was corroded severely, with many deep corrosion pits observed (Figure 13a). For the Mg-2Zn-0.5Gd-0.5Zr alloy, a relatively uniform corrosion was observed, with a few deep corrosion pits (Figure 13b). For the Mg-2Zn-0.5Nd-0.5Zr alloy, it showed a uniform corrosion with few corrosion pits observed, exhibiting the best corrosion resistance (Figure 13c). For the Mg-2Zn-0.5Y-0.5Zr alloy, many corrosion pits could also be observed, with its corrosion resistance only just better than the Mg-2Zn-0.5Zr alloy. Consequently, it can be concluded that the corrosion resistance according to the immersion test is ranked as follows: Mg-2Zn-0.5Nd-0.5Zr alloy > Mg-2Zn-0.5Gd-0.5Zr alloy > Mg-2Zn-0.5Y-0.5Zr alloy > Mg-2Zn-0.5Zr alloy.

### 3.6. Cytotoxicity

Figure 14 shows the cell viability of MC3T3 cells after culturing with extracts of different alloys. It can be found that all the *RGR* values of the Mg-2Zn-0.5Zr alloy at 1, 3 and 5 days were approximately 75–80%. Moreover, the *RGR* values of the Mg-2Zn-0.5Y-0.5Zr, Mg-2Zn-0.5Gd-0.5Zr and Mg-2Zn-0.5Nd-0.5Zr alloys at different culture times were approximately 85%, 90% and 95%, respectively. With an increase in the culture time, the *RGR* values of all of the groups increased. According to the ISO10993-5 standard, all of the alloys exhibited a grade 1 cytotoxicity within the range of 75–99%. Moreover, the Mg-2Zn-0.5Nd-0.5Zr alloy possessed the best cytocompatibility.

## 4. Discussion

### 4.1. Effects of Different Rare Earth Elements on the Mechanical Properties of the ECAPed Alloys

Recently, rare earth elements containing biodegradable Mg alloys have attracted many researchers’ attention due to their high mechanical properties and corrosion resistance. However, few studies that clearly illustrate the different roles of the different rare earth elements in the Mg alloys are available. This part will discuss the effects of different rare earth elements on the mechanical properties of the ECAPed alloys.

The bimodal grain structure played an important role in the mechanical properties [15,16]. Generally speaking, the smaller the grain size, the higher the strength and ductility of the alloy. The large grains are therefore detrimental to the mechanical properties of the alloys. After the four-pass extrusion operation, an obvious bimodal grain structure could be observed. However, this structure was not so evident in the other alloys (Figure 3). For the bimodal grain structure, the larger grains usually cannot coordinate the deformation and cracks then easily emanate from the larger grain boundaries. Therefore, the bimodal grain structure should be avoided. Xiang et al. [17] found that graphene nanoplatelets could reinforce the bimodal structural of the Mg-6Zn alloy by the synergistic strengthening effect of the graphene nanoplatelets and the precipitates. Park et al. [18] found that the addition of elements with a high solid solubility in the α-Mg matrix, such as Sn, could significantly improve the strength of the extruded Mg-Al alloy via enhanced precipitation/grain-boundaries strengthening, and without a loss of ductility. In this study, it was found that the addition of rare earth elements could also decrease the formation of bimodal grain structures. After addition of Gd, Nd and Y, the grain size decreased obviously (Table 3). According to a previous study [19], grain refinement strengthening played an important role in the mechanical properties by increasing the number of ECAP passes. Therefore, the rare earth elements containing Mg alloys exhibited a much higher strength, especially, the Nd and Y elements. It was found that [20] the deformation mechanism in the bimodal grain structure alloy was dominated by basal slips in fine grains and twinning in coarse grains. The twinning deformation needs more energy than the basal slip. Moreover, the twinning deformation is small, which is another factor that can influence the ductility of the Mg-2Zn-0.5Zr alloy. With the addition of rare earth elements, the large grain size decreased, and the deformation changed to a basal slip (Figure 6), leading to the enhancement of the ductility.

Second phases also played vital importance in the mechanical properties after the four-pass extrusion operation. For the Mg-2Zn-0.5Gd-0.5Zr alloy, the main large second phase was the W phase, and the amount of small second phases, mainly the Mg-Zn phase, was low; therefore, they cannot effectively inhibit the movement of the dislocation. Thus, the strength was relatively lower. For the Mg-2Zn-0.5Nd-0.5Zr alloy, the large second phase was the T phase, and the amount of small second phases increased with the obvious enhancement of the second phase strengthening effect. Moreover, it was reported that [21,22] the solid-soluble Nd atoms could activate the non-base slip and decrease the stacking fault energy of Mg alloys. For the Mg-2Zn-0.5Y-0.5Zr alloy, the main large second phase was the W phase. However, the amount of small second phases significantly increased with a uniform distribution. Zn has been found to have an excellent strengthening effect in the Mg alloys, with the highest solubility of 6.2 wt.% [23]. Hence, the combination of second phase strengthening and grain refinement strengthening in the Mg-2Zn-0.5Y-0.5Zr alloy aided its highest strength. 

The mechanism behind the increase in strength after the addition of RE elements can be explained as follows: on one hand, after the addition of the RE elements, the grain refinement effect was improved during the plastic deformation. Moreover, the grain refinement effect was ranked as: Y > Nd > Gd; on the other hand, after the addition of different RE elements, the amount of second phases increased. The second phase strengthening played a vital importance in the enhancement of the strength. Moreover, different strengths were exhibited due to the variations of the second phase distribution. The Y-containing second phases distributed uniformly with a strong dispersion-strengthening effect exhibited the highest strength (Figure 5f). The Gd-containing second phases were much larger than the other two alloys, and the amount was relatively little (Figure 5b). The Mg-2Zn-0.5Gd-0.5Zr alloy exhibited the lowest strength amongst the three kinds of alloys. The strength of the Mg-2Zn-0.5Gd-0.5Zr alloy was, however, in the intermediate.

### 4.2. Effects of Different Rare Earth Elements on the Corrosion Resistance of the ECAPed Alloys

Both plastic deformation and alloying could improve the corrosion resistance of Mg alloys. This part will discuss the effects of different rare earth elements on the corrosion resistance of the ECAPed alloys.

Plastic deformation, especially severe plastic deformation, such as ECAP extrusion, could break the large second phases and make them more uniformly distributed in the Mg alloys [14]. According to a previous study [24], the corrosion rate of Mg alloys is closely related to the microgalvanic effect, which is determined by the second phase and the matrix. The amount of second phases in the four-pass extruded Mg-2Zn-0.5Y-0.5Zr alloy was much larger than that of the Mg-2Zn-0.5Gd-0.5Zr alloy and Mg-2Zn-0.5Nd-0.5Zr alloy (Figure 5), exhibiting a stronger microgalvanic effect. Therefore, the corrosion resistance of the Mg-2Zn-0.5Y-0.5Zr alloy was lower. The second phases in the four-pass extruded Mg-2Zn-0.5Gd-0.5Zr alloy were larger than that of the Mg-2Zn-0.5Nd-0.5Zr alloy. It is well known that the larger second phase is detrimental to the corrosion resistance of alloys. Thus, the corrosion rate of the Mg-2Zn-0.5Gd-0.5Zr alloy was faster than the Mg-2Zn-0.5Nd-0.5Zr alloy. Another factor influencing the corrosion resistance of the Mg-2Zn-0.5Nd-0.5Zr alloy is that Nd could increase the matrix potential and decrease the microgalvanic effect between the matrix and the second phase [25]. After four-pass extrusion, the grain size was decreased obviously. However, a bimodal grain structure formed in the alloys. It was found that the microgalvanic effect also exists between the large grains and the ultrafine grains [3,26]. The surface energy of ultrafine grains is much higher than the large grains. The ultrafine grains acted as an anode and the large grains acted as a cathode. The four-pass extruded Mg-2Zn-0.5Zr alloy showed an obvious bimodal grain structure. After the addition of Gd, Nd and Y, the bimodal grain structure was inhibited to some extent. Therefore, the rare earth element-containing alloys exhibited a much better corrosion resistance than the Mg-2Zn-0.5Zr alloy. Corrosion products on the surface of Mg alloys also played an important role in enhancing their corrosion resistance. Although the corrosion product of the MgO film was not dense, with a low protective effect, it was found that calcium phosphate deposition could protect the matrix [27,28]. From the corrosion product observation (Figure 12), it was discovered that flocculent corrosion products that contained large amounts of Ca and P elements were deposited on the surface of the Mg-2Zn-0.5Nd-0.5Zr alloy. Calcium phosphate deposition also improved the corrosion resistance of the Mg-2Zn-0.5Nd-0.5Zr alloy and exhibited uniform corrosion in Hank’s solution. 

### 4.3. Effects of Different Rare Earth Elements on the Cytocompatibility of the ECAPed Alloys

It has been proven that Mg alloys have good biocompatibility, and some nutritious alloying elements, such as Ca, Sr, Si, Mn, etc., have also proven to be non-toxic. However, the biocompatibility of the rare earth elements still remains a subject of debate. Feyerabend et al. [29] found that the cytotoxicity of the rare earth elements may be related to their ionic radii. For the highly soluble elements, Dy and Gd are more suitable than Y. Suitable elements with a low solid solubility are Nd, Eu and Pr. In this study, it was found that Nd had the best cytocompatibility and Y exhibited the lowest cytocompatibility. For Mg alloys, the cytocompatibility is closely related to their degradation rate, where a faster degradation rate means a lower cytocompatibility. After the four-pass extrusion, the Mg-2Zn-0.5Nd-0.5Zr alloy showed the lowest corrosion rate, with a degradation rate of 0.42 mm/y in Hank’s solution after 14 days of immersion. In addition, a previous study on the Mg-Nd-Zn-Zr alloy also indicated that Nd had good cytocompatibility [30]. It has been proven that Nd^3+^ is beneficial to the differentiation and formation of mineralized matrix nodules of osteoclasts at concentration of 1 × 10^−8^ mol/L and 1 × 10^−5^ mol/L, respectively [31].Therefore, the Mg-2Zn-0.5Nd-0.5Zr alloy showed the highest cell viability. Although there is still controversy about the biosafety of Y, it was found that the Mg-2Zn-0.5Y-0.5Zr alloy has a relative low cytocompatibility in this work. For the Mg-2Zn-0.5Y-0.5Zr alloy, it was easily corroded due to the microgalvanic effect and low Gibbs free energy of Y, where a faster degradation would produce more Mg^2+^ ions and a higher pH value, thus decreasing the cell viability of the alloys. A further study was carried out in order to analyze the cytocompatibility of Y^3+^, and it was found that the effects of Y^3+^ on the proliferation, differentiation and mineralization functions depended on the concentration and culture time [32]. For the Mg-2Zn-0.5Gd-0.5Zr alloy, though it exhibited a good cytocompatibility, there was no metabolic pathway for the Gd element in vivo that would accumulate in the organs of rats after implantation [33]. Thus, Gd-free WE43 (MgYREZr) alloys are being explored and applied in clinics [34], and, in this sense, the Mg-2Zn-0.5Nd-0.5Zr alloy shows the best potential to be used in the field of biomaterials.

## 5. Conclusions

In this study, the effects of different rare earth elements Gd, Nd and Y on the degradation and mechanical properties of the ECAP extruded Mg-2Zn-xRE-0.5Zr alloys were investigated. Some conclusions that can be drawn are as follows:

(1)After the ECAP extrusion, the Mg-2Zn-0.5Y-0.5Zr alloy shows the highest strength due to the second phase dispersive strengthening. Moreover, the Mg-2Zn-0.5Nd-0.5Zr alloy exhibits the best ductility due to the grain refinement and activation of the non-basal slip. This means that both Y and Nd are beneficial to the enhancement of strength and ductility, respectively;(2)The Mg-2Zn-0.5Nd-0.5Zr alloy shows the best corrosion resistance, exhibiting a uniform corrosion; the Mg-2Zn-0.5Y-0.5Zr alloy exhibits the lowest corrosion resistance, revealing a pitting corrosion. The corrosion resistance of the alloys by the addition of the three elements is ranked as follows: Nd > Gd > Y;(3)The three added elements (i.e., Nd, Gd and Y) possess good cytocompatibility. Meanwhile, combining the three parameters of the mechanical property, corrosion resistance and cytocompatibility, the ECAPed Mg-2Zn-0.5Nd-0.5Zr alloy shows a good application prospect in the field of orthopedics.

## Figures and Tables

**Figure 1 materials-15-00627-f001:**
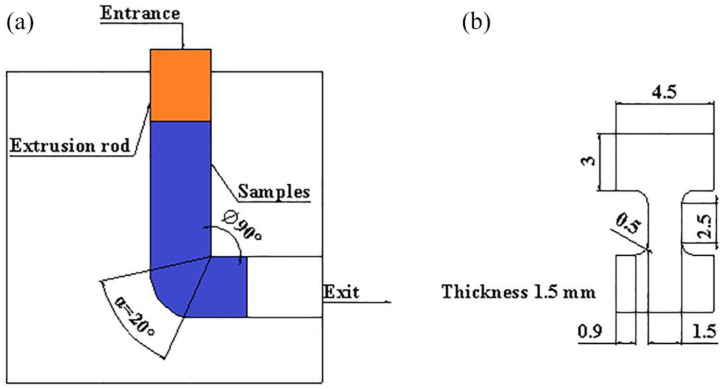
Schematic illustration of the extrusion mold and the tensile samples: (**a**) extrusion mold; (**b**) tensile sample.

**Figure 2 materials-15-00627-f002:**
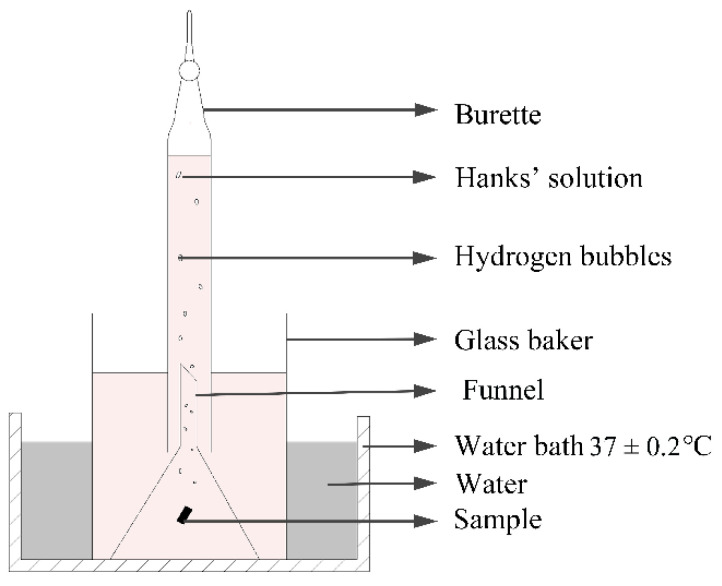
Schematic diagram of hydrogen evolution installation.

**Figure 3 materials-15-00627-f003:**
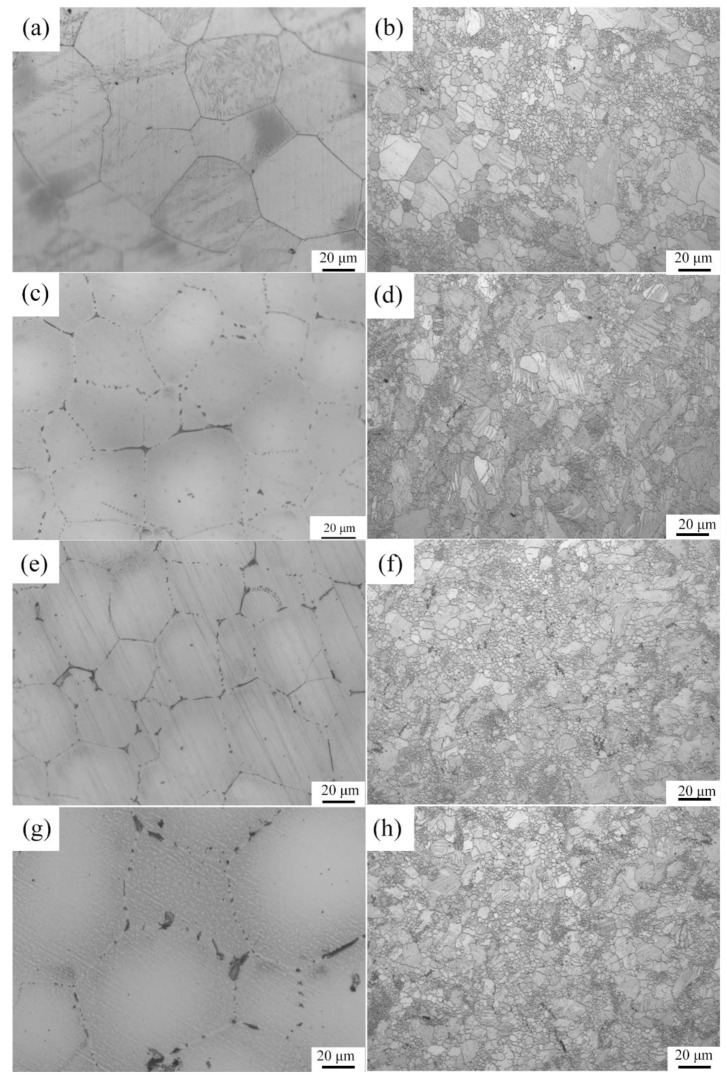
Optical microstructure of the alloys: (**a**,**b**) the as-cast and the four-pass extruded Mg-2Zn-0.5Zr alloy; (**c**,**d**) the as-cast and the four-pass extruded Mg-2Zn-0.5Gd-0.5Zr alloy; (**e**,**f**) the as-cast and the four-pass extruded Mg-2Zn-0.5Nd-0.5Zr alloy; (**g**,**h**) the as-cast and the four-pass extruded Mg-2Zn-0.5Y-0.5Zr alloy.

**Figure 4 materials-15-00627-f004:**
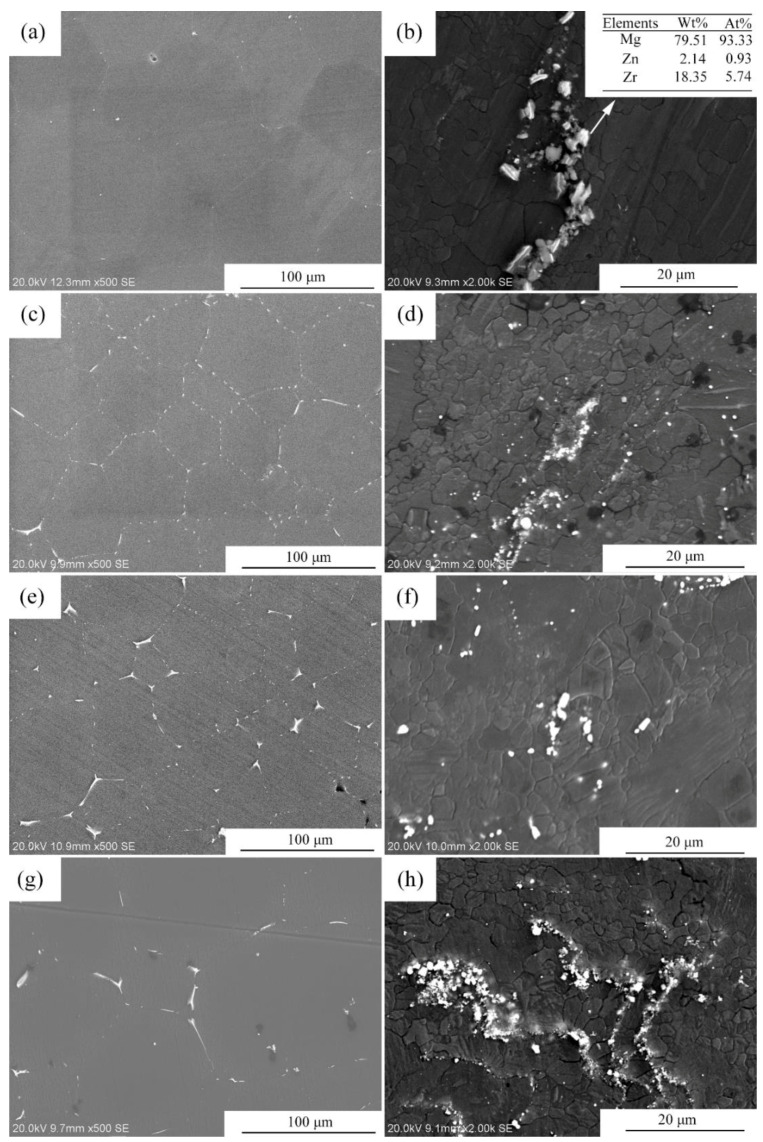
Microstructures of the four-pass extruded alloys: (**a**,**b**) the as-cast and the four-pass extruded Mg-2Zn-0.5Zr alloy; (**c**,**d**) the as-cast and the four-pass extruded Mg-2Zn-0.5Gd-0.5Zr alloy; (**e**,**f**) the as-cast and the four-pass extruded Mg-2Zn-0.5Nd-0.5Zr alloy; (**g**,**h**) the as-cast and the four-pass extruded Mg-2Zn-0.5Y-0.5Zr alloy.

**Figure 5 materials-15-00627-f005:**
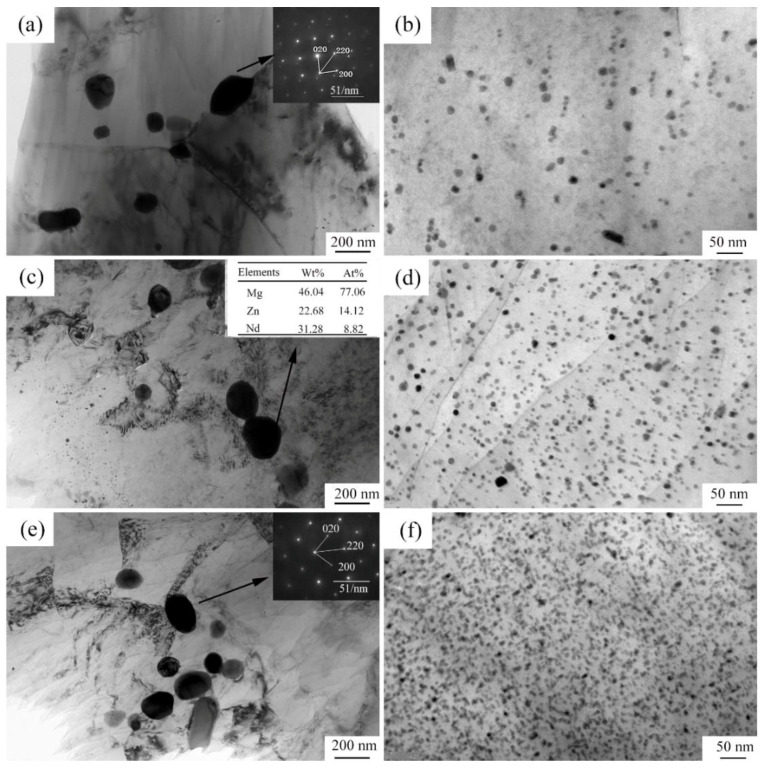
TEM images of the four-pass extruded alloys: (**a**,**b**) Mg-2Zn-0.5Gd-0.5Zr alloy; (**c**,**d**) Mg-2Zn-0.5Nd-0.5Zr alloy; (**e**,**f**) Mg-2Zn-0.5Y-0.5Zr alloy.

**Figure 6 materials-15-00627-f006:**
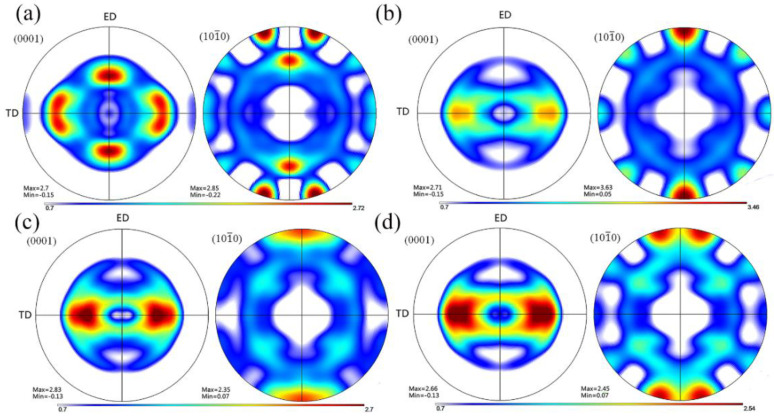
Textures of the four-pass extruded alloys: (**a**) Mg-2Zn-0.5Zr alloy [13]; (**b**) Mg-2Zn-0.5Gd-0.5Zr alloy [13]; (**c**) Mg-2Zn-0.5Nd-0.5Zr alloy; and (**d**) Mg-2Zn-0.5Y-0.5Zr alloy.

**Figure 7 materials-15-00627-f007:**
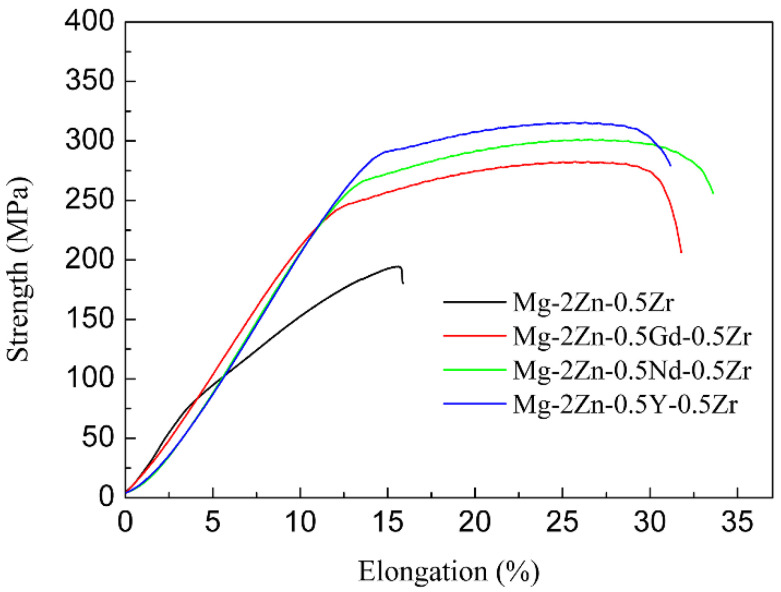
Stress-strain curves of the alloys after the four-pass extrusion.

**Figure 8 materials-15-00627-f008:**
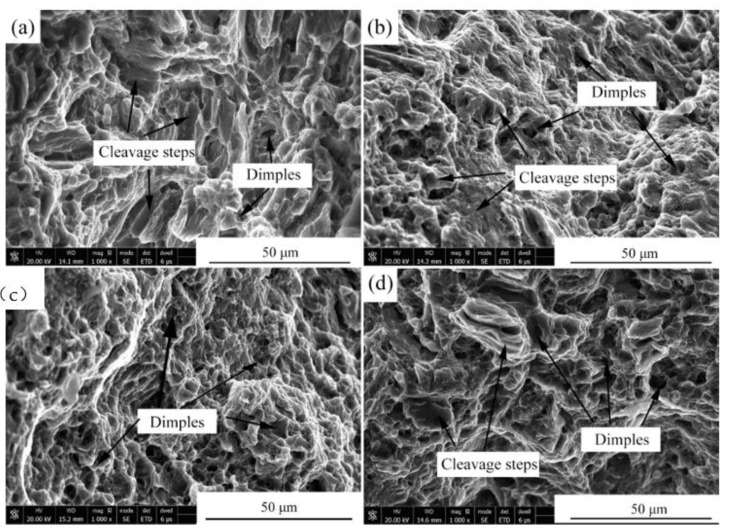
Fracture morphologies of the four-pass extrusion alloys: (**a**) Mg-2Zn-0.5Zr alloy; (**b**) Mg-2Zn-0.5Gd-0.5Zr alloy; (**c**) Mg-2Zn-0.5Nd-0.5Zr alloy; and (**d**) Mg-2Zn-0.5Y-0.5Zr alloy.

**Figure 9 materials-15-00627-f009:**
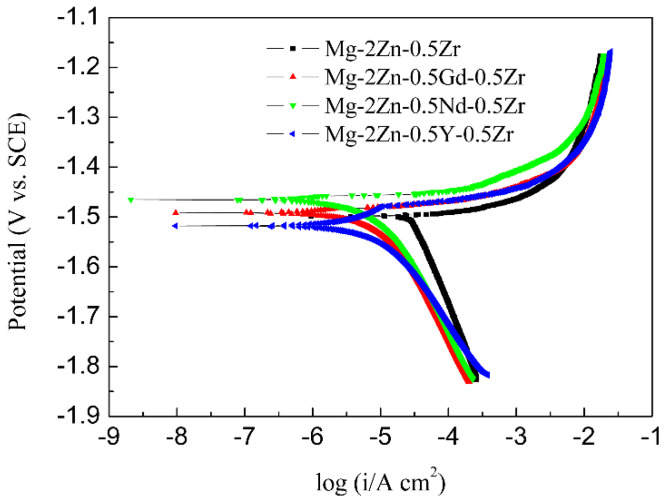
Potentiodynamic polarization curves of the alloys in Hank’s solution at 37 °C.

**Figure 10 materials-15-00627-f010:**
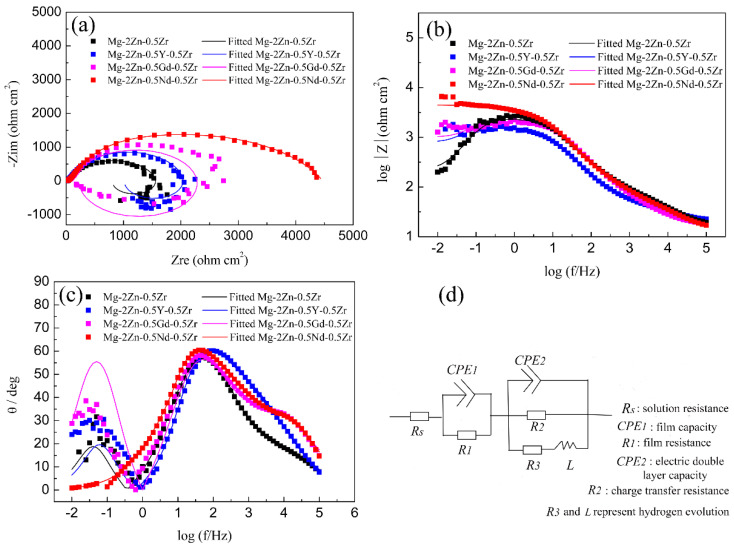
EIS curves of the alloys in Hank’s solution: (**a**) Nyquist curves; (**b**) Bode curves; (**c**) phase angle curves; and (**d**) equivalent circuit.

**Figure 11 materials-15-00627-f011:**
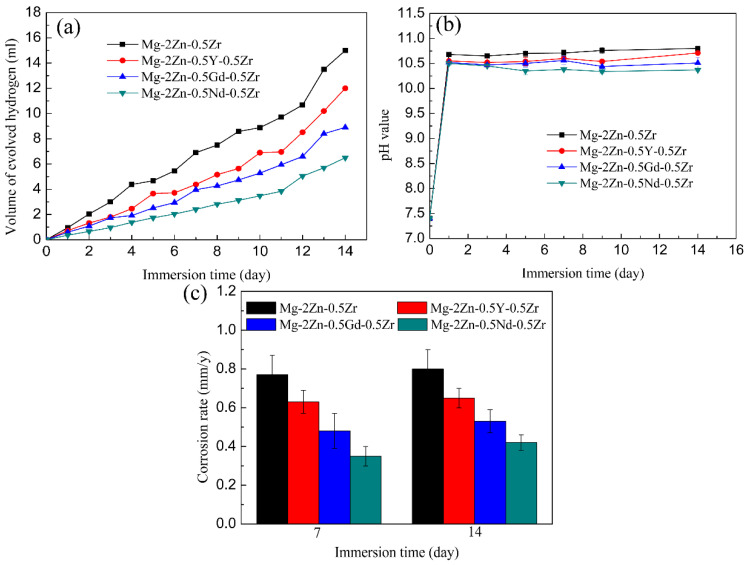
(**a**) Hydrogen evolution of the samples in Hank’s solution; (**b**) pH value of the immersion solution; (**c**) corrosion rate after 7 days and 14 days of immersion.

**Figure 12 materials-15-00627-f012:**
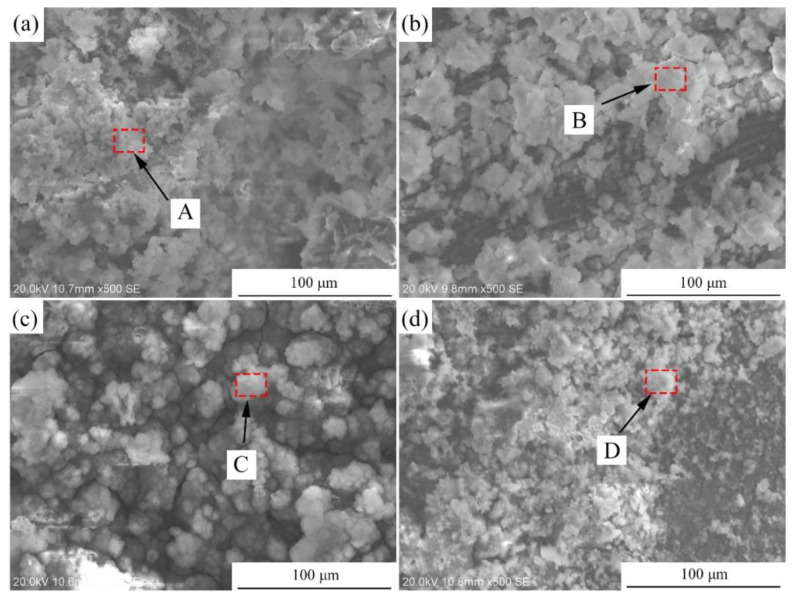
Corrosion morphologies of the four-pass extrusion alloys after 14 days of immersion in Hank’s solution with corrosion products: (**a**) Mg-2Zn-0.5Zr alloy; (**b**) Mg-2Zn-0.5Gd-0.5Zr alloy; (**c**) Mg-2Zn-0.5Nd-0.5Zr alloy; and (**d**) Mg-2Zn-0.5Y-0.5Zr alloy.

**Figure 13 materials-15-00627-f013:**
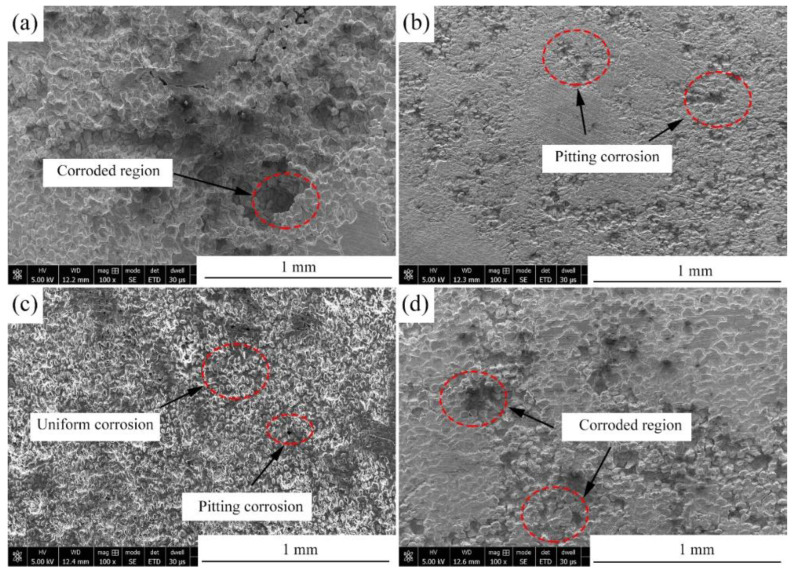
Corrosion morphologies of the four-pass extrusion alloys after 14 days of immersion in Hank’s solution with removal of the corrosion products: (**a**) Mg-2Zn-0.5Zr alloy; (**b**) Mg-2Zn-0.5Gd-0.5Zr alloy; (**c**) Mg-2Zn-0.5Nd-0.5Zr alloy; and (**d**) Mg-2Zn-0.5Y-0.5Zr alloy.

**Figure 14 materials-15-00627-f014:**
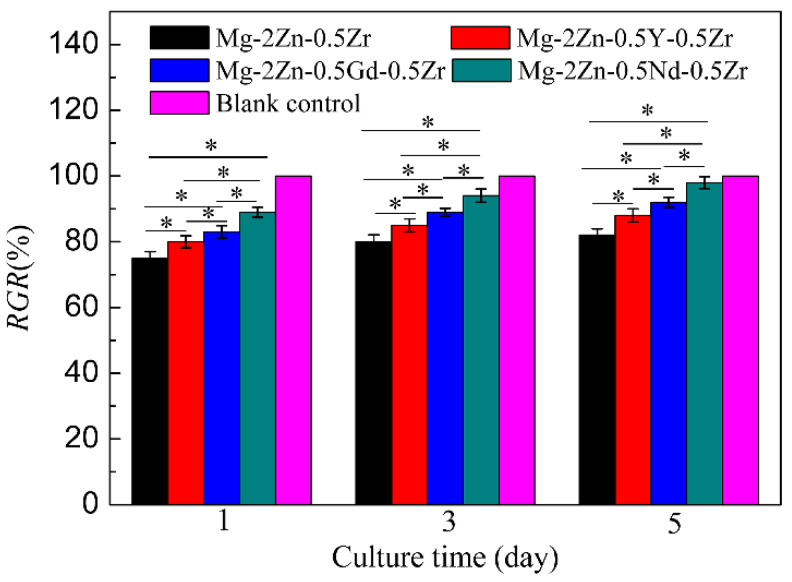
Cell viability of MC3T3-E1 cells after culturing with extracts of different alloys for 1, 3 and 5 days (* *p* < 0.05).

**Table 1 materials-15-00627-t001:** Actual composition of the three alloys.

Alloys	Composition (wt.%)					
	Zn	Gd	Nd	Y	Zr	Mg
Mg-2Zn-0.5Gd-0.5Zr	2.00	0.53	-	-	0.48	Bal.
Mg-2Zn-0.5Nd-0.5Zr	2.23	-	0.51	-	0.49	Bal.
Mg-2Z-0.5Y-0.5Zr	2.07	-	-	0.51	0.38	Bal.

**Table 2 materials-15-00627-t002:** Chemical composition of Hank’s solution (g/L).

Composition	NaCl	KCl	KH_2_PO_4_	MgSO_4_	NaHCO_3_	CaCl_2_	Na_2_HPO_4_	Glucose
Content (g/L)	8.00	0.40	0.06	0.20	0.35	0.14	0.12	1.00

**Table 3 materials-15-00627-t003:** Grain size of the alloys before and after extrusion.

Alloys	Grain Size (μm)	
	As-Cast	After Four-Pass Extrusion
Mg-2Zn-0.5Zr	52.06 ± 7.62	4.99 ± 2.20
Mg-2Zn-0.5Gd-0.5Zr	47.48 ± 11.50	4.40 ± 0.86
Mg-2Zn-0.5Nd-0.5Zr	38.84 ± 5.44	3.95 ± 0.26
Mg-2Zn-0.5Y-0.5Zr	63.93 ± 5.22	2.84 ± 0.32

**Table 4 materials-15-00627-t004:** The Gibbs free energy calculated by Factsage software.

Reaction	Gibbs Free Energy ΔG(J)/298.15 K	Gibbs Free Energy ΔG(J)/1273.15 K
$2Y+\frac{3}{2}O_{2}\to Y_{2}O_{3}$	−3,639,103.7	−3,075,633.2
$2\mathrm{Gd}+\frac{3}{2}O_{2}\to Gd_{2}O_{3}$	−3,468,204.2	−2,924,028.9
$2\mathrm{Nd}+\frac{3}{2}O_{2}\to Nd_{2}O_{3}$	−3,437,913.5	−2,900,746.2

**Table 5 materials-15-00627-t005:** Mechanical properties of the alloys after the four-pass extrusion.

Alloys	UTS (MPa)	YS (MPa)	EL(%)
Mg-2Zn-0.5Zr	194 ± 2	104 ± 3	10 ± 1
Mg-2Zn-0.5Gd-0.5Zr	280 ± 3	250 ± 2	19 ± 1
Mg-2Zn-0.5Nd-0.5Zr	300 ± 4	270 ± 3	22 ± 2
Mg-2Zn-0.5Y-0.5Zr	315 ± 3	295 ± 2	17 ± 1

**Table 6 materials-15-00627-t006:** Tafel fitting results of the potentiodynamic polarization curves of the alloys.

Samples	*E* (V. vs. SCE)	*i_corr_* (μA/cm^2^)	*CR* (mm/year)
Mg-2Zn-0.5Zr	−1.50 ± 0.01	30.46 ± 2.50	0.70 ± 0.06
Mg-2Zn-0.5Gd-0.5Zr	−1.49 ± 0.02	12.02 ± 1.05	0.27 ± 0.02
Mg-2Zn-0.5Nd-0.5Zr	−1.47 ± 0.01	10.38 ± 0.85	0.24 ± 0.02
Mg-2Zn-0.5Y-0.5Zr	−1.52 ± 0.01	15.10 ± 2.00	0.35 ± 0.05

**Table 7 materials-15-00627-t007:** Fitting results of the EIS curves based on the equivalent circuit.

Samples	*R_s_* (Ω cm^2^)	*CPE* _1_		*R*_1_ (Ω cm^2^)	*CPE* _2_		*R*_2_ (Ω cm^2^)	*R*_3_ (Ω cm^2^)	*L* (H cm^−2^)
		*Y*_01_ (S·sec^^^n/cm^2^)	n_1_		*Y*_02_ (S·sec^^^n/cm^2^)	n_2_			
Mg-2Zn-0.5Zr	19.8	9.2 × 10^−5^	0.6	34.7	2.8 × 10^−5^	0.8	1.5 × 10^3^	1.4 × 10^3^	8.1 × 10^3^
Mg-2Zn-0.5Gd-0.5Zr	14.2	5.1 × 10^−5^	0.6	105.6	1.4 × 10^−5^	0.9	2.2 × 10^3^	110.8	2.3 × 10^3^
Mg-2Zn-0.5Nd-0.5Zr	13.8	2.2 × 10^−5^	0.7	25.7	3.1 × 10^−5^	0.7	2.8 × 10^5^	4.5 × 10^3^	57.64
Mg-2Zn-0.5Y-0.5Zr	16.6	1.0 × 10^−^^4^	0.6	93.42	1.0 × 10^−^^5^	0.9	2.0 × 10^3^	1.7 × 10^3^	6.6 × 10^3^

**Table 8 materials-15-00627-t008:** Chemical composition of the areas marked in Figure 10.

Position	Chemical Composition (wt.%)				
	C	O	Mg	P	Ca
A	8.66	53.58	40.00	3.61	3.15
B	5.21	54.34	36.46	2.38	1.58
C	9.61	49.22	21.75	9.98	9.44
D	8.93	51.09	29.06	5.52	5.40

## Data Availability

The datasets generated during and/or analysed during the current study are available from the corresponding author on reasonable request.

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
