# Peer review of "Effects of Different Rare Earth Elements on the Degradation and Mechanical Properties of the ECAP Extruded Mg Alloys"

_materials, 2022, doi:10.3390/ma15020627_

Round 1
Reviewer 1 Report
The manuscript entitled: Effects of different rare earth elements on the degradation and mechanical properties of the ECAP extruded Mg alloys deals with the influence of rare earth elements like Y, Nd, Gd, etc. on the properties of ECAPed Mg. I have the following concerns with the present manuscript.
- The choice of three elements Y, Nd, and Gd is still not justified. Why other RE elements are not considered and only these three?
- XRD patterns of these samples should be introduced to give detailed information about the phases present in these samples before and after extrusion. In case, if there are no phase changes, information such as crystallite size, lattice strain, and dislocation density could be extracted to have meaningful information about the samples at different stages. Without XRD patterns discussing the phases present like W Phase, T phase, etc. is questionable, since there is no experimental evidence to document the same.
- A scientific explanation of the findings is missing.
- I do not think Fig. 5(b,d,f) gives any useful information within the context of the present manuscript.
- What is the mechanism behind the increase in strength after the addition of RE elements?
- Typos in the manuscript should be carefully rectified. For instance, 2wt% should be written as 2 wt.%
- The English language needs attention.
Author Response
The choice of three elements Y, Nd, and Gd is still not justified. Why other RE elements are not considered and only these three?
Reply: Dear Professor, thank you for your insightful question. However, as can be seen in the work of Chen et al., the respective solubilities of the Nd, Y and Gd in Mg alloys are 3.6 wt.%, 12.4 wt.% and 23.49 wt.% (Recent advances on the development of magnesium alloys for biodegradable implants, 10.1016/j.actbio.2014.07.005). They represent the different kinds of RE-containing Mg alloys with different solubility. These three different elements were thus chosen in this study. In addition, these three elements contained in alloys have been severally studied, such as WE43 alloy, Mg-Nd-Zn-Zr alloy and Mg-Gd-Zn-Zr alloy (Microstructures and corrosion behavior of biodegradable Mg-6Gd-xZn-0.4Zr alloys with and without long period stacking ordered structure, 10.1016/j.corsci.2016.01.004).
XRD patterns of these samples should be introduced to give detailed information about the phases present in these samples before and after extrusion. In case, if there are no phase changes, information such as crystallite size, lattice strain, and dislocation density could be extracted to have meaningful information about the samples at different stages. Without XRD patterns discussing the phases present like W Phase, T phase, etc. is questionable, since there is no experimental evidence to document the same.
Reply: Dear Professor, thank you for your suggestion. According to our previous studies, no phase changes in the alloys occurred, when analyzed by XRD during the plastic deformation. The relative studies are: [1] Chen J, Tan L, Yu X, Yang K. Effect of minor content of Gd on the mechanical and degradable properties of as-cast Mg-2Zn-xGd-0.5Zr alloys[J]. Journal of Materials Science & Technology, 2019, 35: 503-511.
[2] Chen J, Tan L, Etim IP, Yang K. Comparative study of the effect of Nd and Y content on the mechanical and biodegradable properties of Mg-Zn-Zr-xNd/Y (x= 0.5, 1, 2) alloys[J]. Materials Technology, 2018, 33: 659-671.
[3] Chen J, Gao M, Tan L, Yang K. Microstructure, mechanical and biodegradable properties of a Mg-2Zn-1Gd-0.5Zr alloy with different solution treatments[J]. Rare Metals, 2019, 6(38): 532-542.
A scientific explanation of the findings is missing.
Reply: Dear Professor, the scientific explanation has now been added in Part 4.1. We adopted your suggestion and discussed the mechanism behind the increase in strength after the addition of RE elements. Thank you for your useful suggestion.
I do not think Fig. 5(b,d,f) gives any useful information within the context of the present manuscript.
Reply: Dear Professor, we have discussed Fig. 5(b,d,f) in Part 4.1. The difference in strength of the three alloys was because of the microstructural difference.
What is the mechanism behind the increase in strength after the addition of RE elements?
Reply: The mechanism behind the increase in strength after the addition of RE elements can be explained as follows: on one hand, after addition of the RE elements, the grain refinement effect was improved during the plastic deformation. Moreover, the grain refinement effect was ranked as: Y > Nd > Gd; on the other hand, after addition of different RE elements, the amount of second phases increased. The second phase strengthening played a vital importance in the enhancement of the strength. Moreover, the difference in strength exhibited was due to the variations in the second phases distribution. The Y-containing second phases was distributed uniformly with strong dispersion-strengthening effect exhibiting the highest strength (Fig. 5f). The Gd-containing second phases were much larger than the other two alloys and the amount was relatively little (Fig. 5b). The Mg-2Zn-0.5Gd-0.5Zr alloy exhibited the lowest strength amongst the three kinds of alloys. The strength of Mg-2Zn-0.5Gd-0.5Zr alloy was in the intermediate.
Typos in the manuscript should be carefully rectified. For instance, 2wt% should be written as 2 wt.%.
Reply: These typos have been double-checked and revised accordingly. Thank you.
The English language needs attention.
Reply: Dear Professor, a native English speaker has been invited to further revise the language. Thank you.

Reviewer 2 Report
The paper by J. Chen, K. Yang et al. reports an investigation about the influence of three rare earth elements on the degradation and mechanical properties of a Mg alloy subjected to ECAP extrusion. Using several techniques, the authors show the improvements that the addition of about 0.5% of each element brings to the alloy in terms of strength and corrosion resistance. Furthermore, higher biocompatibility is demonstrated by culturing murine cells with extract of alloys containing rare earth elements with respect to the pristine alloy.
The paper is generally well written, good quality data are shown and the discussion of the results is exhaustive. I suggest to publish the paper, by inviting the authors to consider the following, minor remarks:
- Better avoid acronyms in the abstract (UTS, YS).
- Figure 2 contains a typo: ‘funeel’, istead of funnel.
- page 8: please, explain the acronym MTT.
- Table 4: a reference to the Factsage software should be given.
- page 30, one line above the Conclusion: ‘in this wise’. There has to be a typo, why ‘wise’?
- reference 27: the pages of the journal are missing.
Author Response
- Better avoid acronyms in the abstract (UTS, YS).
Reply: Dear professor, we have revised it in the abstract.
- Figure 2 contains a typo: ‘funeel’, istead of funnel.
Reply: Dear professor, thank you for your careful revision. It was revised in Fig.2.
- page 8: please, explain the acronym MTT.
Reply: MTT(3-(4,5)-dimethylthiahiazo(-z-yl)-3,5-diphenytetrazoliumromide)
- Table 4: a reference to the Factsage software should be given.
Reply: Dear professor, I have checked it again. Factsage 8.0 was used in this experiment.
- page 30, one line above the Conclusion: ‘in this wise’. There has to be a typo, why ‘wise’?
Reply: Dear professor, thank you for your suggestion. We have revised it. “and in this sense, Mg-2Zn-0.5Nd-0.5Zr alloy shows the best potential to be used in the field of biomaterials.”
- reference 27: the pages of the journal are missing.
Reply: the DOI of the reference 27 has been added. doi:10.1002/adem.201800121

Round 2
Reviewer 1 Report
The authors have satisfactorily addressed my comments and the manuscript may now be accepted for publication in the present form.